# Mid-Term Estimates of Influenza Vaccine Effectiveness against the A(H1N1)pdm09 Prevalent Circulating Subtype in the 2023/24 Season: Data from the Sicilian RespiVirNet Surveillance System

**DOI:** 10.3390/vaccines12030305

**Published:** 2024-03-14

**Authors:** Claudio Costantino, Walter Mazzucco, Giorgio Graziano, Carmelo Massimo Maida, Francesco Vitale, Fabio Tramuto

**Affiliations:** 1Department of Health Promotion Sciences, Maternal and Infant Care, Internal Medicine and Excellence Specialties “G. D’Alessandro”, University of Palermo, 90127 Palermo, Italy; walter.mazzucco@unipa.it (W.M.); carmelo.maida@unipa.it (C.M.M.); francesco.vitale@unipa.it (F.V.); fabio.tramuto@unipa.it (F.T.); 2Sicilian Reference Laboratory for Integrated Surveillance of Respiratory Viruses, University Hospital of Palermo, 90127 Palermo, Italy; giorgio.graziano@unipa.it

**Keywords:** influenza epidemic, A(H1N1)pdm09, vaccine effectiveness, virological surveillance, test-negative design

## Abstract

The current influenza season started in Italy in October 2023, approaching the epidemic peak in late December (52nd week of the year). We aimed to explore the mid-term virologic surveillance data of the 2023/2024 influenza season (from 16 October 2023 to 7 January 2024) in Sicily, the fourth most populous Italian region. A test-negative design was used to estimate the effectiveness of seasonal influenza vaccine (VE) against A(H1N1)pdm09 virus, the predominant subtype in Sicily (96.2% of laboratory-confirmed influenza cases). Overall, 29.2% (*n* = 359/1230) of oropharyngeal swabs collected from patients with influenza-like illness (ILI) were positive for influenza. Among the laboratory-confirmed influenza cases, 12.5% (*n* = 45/359) were vaccinated against influenza, with higher prevalence of laboratory-confirmed diagnosis of influenza A among subjects vaccinated with quadrivalent inactivated standard dose (29.4%), live attenuated intranasal (25.1%), and quadrivalent inactivated high-dose (23.8%) influenza vaccines. Comparing influenza vaccination status for the 2023/2024 season among laboratory-confirmed influenza-positive and -negative samples, higher vaccination rates in influenza-negative samples (vs. positive) were observed in all age groups, except for 45–64 years old, regardless of sex and comorbidities. The overall adjusted VE (adj-VE) was 41.4% [95%CI: 10.5–61.6%], whereas the adj-VE was 37.9% [95%CI: −0.7–61.7%] among children 7 months–14 years old and 52.7% [95%CI: −38.0–83.8%] among the elderly (≥65 years old).

## 1. Introduction

Starting from February 2022, influenza virus circulation returned to pre-COVID-19 pandemic levels worldwide, with A(H3N2), A(H1N1)pdm09, and B viruses being detected in most countries [1].

Therefore, the importance of sustaining ongoing global influenza surveillance has re-emerged, as well as the evaluation of vaccine effectiveness and the promotion of influenza vaccination, in order to minimize the potential public health consequences of seasonal influenza spread [2,3,4].

The recommended composition provided by the World Health Organization for the 2023/2024 season vaccines in the Northern Hemisphere included (a) the A/Victoria/4897/2022 (H1N1)pdm09-like virus (for egg-based vaccines) or the A/Wisconsin/67/2022 (H1N1)pdm09-like virus (both drifted during last season), (b) the A/Darwin/9/2021 (H3N2)-like virus, and (c) the B/Austria/1359417/2021 (B/Victoria lineage)-like virus and B/Phuket/3073/2013 (B/Yamagata lineage)-like virus [5].

The 2023/2024 influenza season in Europe started earlier than previous ones, at the end of November 2023 [6]. In Italy, the epidemiological trend of influenza-like illness (ILI) was anticipated as compared to the past 19 cold seasons (except for the 2009/2010 A(H1N1) pandemic season), with a peak incidence value observed from the 48th to the 51st week of 2023 and the highest value reported so far [7].

Several other respiratory viruses contributed to the incidence of ILIs in Italy, mainly including SARS-CoV-2, human respiratory syncytial viruses (hRSVs) A and B, adenovirus, rhinoviruses, and metapneumovirus, all included in the national respiratory virus surveillance network (RespiVirNet) protocol [8].

As a result of the early start of influenza season in 2023/2024, virological and epidemiological surveillance was activated in Italy starting from October 2023, approaching the epidemic peak across the end of December 2023 [7]. At the mid-term of the 2023/2024 season, in this country, A(H1N1)pdm09 resulted in being the highly prevalent influenza virus (77.7%) [9].

In Sicily, both quadrivalent inactivated (QIV) and live attenuated (LAIV) influenza vaccines were offered to the general population according to the guidelines provided by the Italian Ministry of Health, which recommend immunization for all subjects older than 6 months of age without contraindications to the vaccine [10].

In more detail, a strong recommendation and an active offer of seasonal influenza vaccination were provided to individuals aged ≥65 years old, those older than 6 months of age at high-risk due to severe comorbidities, pregnant women, healthcare professionals, and healthy children attending schools from 6 months to 6 years of age [10].

During the last influenza seasons, the high value of the surveillance mid-term estimates of influenza vaccine effectiveness (VE) reports was demonstrated [2,3,4]. These data, as a matter of fact, can help International Public Health Authorities in determining the efficacy of vaccination programs or track trends in infection cases among the general population in the Northern and Southern Hemispheres [3,4]. Herein, we reported the mid-term vaccine effectiveness (VE) estimates for the 2023/2024 influenza season against the prevalent A(H1N1)pdm09 subtype, provided by the Sicilian RespiVirNet regional reference laboratory.

## 2. Materials and Methods

Since the 2009 influenza pandemic, the regional reference laboratory for influenza surveillance in Sicily, operating at the University Hospital (UH) of Palermo, has been supporting the Italian RespiVirNet network (formerly the InfluNet until 2022/2023 season) [9], providing the epidemiological, virologic, and molecular surveillance of the fourth most populous Italian region, accounting for about 4.8 million resident inhabitants [11].

Starting from the 42nd week of 2023 (16 October 2023), oropharyngeal samples of subjects with ILI were collected by a representative network of sentinel general practitioners (GPs) and family pediatricians (FPs), covering at least 4% of the Sicilian general population, as required by the Ministry of Health [9].

Participation to the RespiVirNet National Surveillance system of sentinel GPs and FPs is on a voluntary basis [9]. Specifically, in the Sicilian region, a central selection of volunteers is annually conducted (substituting GPs or FPs that have not worked well during the previous season or who retired) by the representatives of the regional sections of the scientific societies of general practitioners and family pediatricians [9].

All specimens were molecularly analyzed through real-time PCR-based protocols for influenza type A and B viruses, SARS-CoV-2, hRSV-A and B, rhinovirus, adenovirus, and metapneumovirus (protocols available upon request) [12,13,14].

Any person presenting a sudden onset of symptoms, including at least one systemic symptom such as fever or feverishness, malaise, headache, and myalgia, and concomitantly with at least one respiratory symptom such as cough, sore throat, and shortness of breath, met the inclusion criteria for ILI and were then included in the surveillance, in accordance with the European Commission Decision of June 2018 [15,16].

Socio-demographic and clinical data, vaccination status against influenza, and type of vaccines administered were also collected by sentinel GPs and FPs.

The characteristics of the quadrivalent influenza vaccines (QIVs) offered in Sicily, according to the recommendations for the 2023/2024 season (Italian and Sicilian Region Decrees for Prevention and Control of Influenza), are reported in Table 1 [10,17].

Similarly to previous studies conducted on influenza VE estimates in the Sicilian region, all subjects vaccinated less than 2 weeks before the onset of ILI symptoms, newborns younger than 6 months, and children younger than 9 years old not compliant with the two-dose recommendation for first influenza vaccination were excluded from the analysis (*n* = 8, 0.65%) [2,3].

### Statistical Analysis

The test-negative design, commonly used as the preferred method in observational studies on vaccines, is frequently used to assess influenza vaccine effectiveness (VE) seasonally [2,18].

Moreover, the present design represents the preferred and standardized method used for evaluating the effectiveness of vaccines in real-world settings and has been widely used in influenza vaccine effectiveness studies [16,19].

In a test-negative design study regarding influenza vaccination VE, patients seeking medical care for acute respiratory illness are tested for the disease in question, in accordance with the ILI definition of the European Commission. 

Those who tested positive for the virus in the present study, mainly the A(H1N1)pdm09 virus, were classified as cases, and those who tested negative served as controls [2,3]. 

This design allows researchers to compare the vaccination status of cases (those who tested positive for the virus) with that of controls (those who tested negative) to estimate the vaccine’s effectiveness. 

To this end, among subjects with ILI symptoms, cases were identified as patients with a laboratory-confirmed diagnosis of influenza, and controls comprised individuals without a laboratory-confirmed diagnosis of influenza [3,16,18].

All subjects with an interval between the onset of symptoms and oropharyngeal specimen collection ≥8 days (*n* = 7, 0.57%) were further excluded from the analysis [2,3,18]. The VE estimates were obtained by comparing the odds ratio (ORs) of vaccination between cases and controls (1 − OR × 100) [18].

A logistic regression model was applied to estimate crude and adjusted VE (adj-VE), by sex and the presence of at least one comorbidity, in the overall population and those stratified by age groups (7 months–14 years and ≥65 years old), for the A(H1N1)pdm09 subtype, the predominant influenza A subtype in 2023/2024.

The approval of the Ethical Committee of the University Hospital of Palermo was obtained in November 2023 (protocol 9 of 2023, 6 November 2023).

## 3. Results

### Surveillance Data

Between 16 October 2023 and 7 January 2024, a total of 1230 samples were collected by sentinel GPs and FPs following a community-based approach (Table 2). Overall, 359 (29.2%) samples resulted in being positive for influenza viruses, with a peak observed in the first week of 2024 (Figure 1). Among laboratory-confirmed cases, 45 (12.5%) were vaccinated against influenza.

Almost all influenza infections were sustained by influenza type A viruses, of which 96.2% (*n* = 345) were caused by the A (H1N1)pdm09 subtype (Table 2).

Comparing the data of the 2023/2024 season with the available previous ones (from 2015/2016 to 2022/2023), a prevalence of laboratory-confirmed influenza samples among vaccinated subjects higher than 6% was reported for two seasons only (2016/2017 and 2018/2019 with 9.0% and 9.2% of cases, respectively), which were characterized by a considerable higher circulation of A(H3N2) influenza virus (93.4% and 62.2%, respectively), which is different from the current one (Table 2).

Table 3 reports the distribution of the laboratory-confirmed influenza cases among vaccinated subjects according to the influenza vaccine type received during the 2023/2024 season. Among vaccinated subjects (*n* = 191), regardless of vaccine type received, 23.6% (*n* = 45) had a laboratory-confirmed diagnosis of influenza (Table 3).

In comparison with the average prevalence observed among vaccinated subjects, a slightly higher prevalence of laboratory-confirmed cases was reported among individuals vaccinated with QIVsd (29.4%), LAIV (25.4%), an QIVhd (23.1%) vaccines, while lower frequencies were found among subjects vaccinated with QIVcc (11.1%) and QIVa (11.8%) (Table 3).

Table 4 reports the main characteristics of the 1230 individuals with ILI monitored by the Sicilian RespiVirNet surveillance system during the current influenza season. Most of them (*n* = 798; 64.9%) were ≤15 years old, followed by 284 subjects (23.1%) aged 15–64 years, and 148 (12.0%) were ≥65 years old. About half of the subjects (*n* = 625; 50.8%) were female, whereas 14.2% (*n* = 175) had at least one comorbidity.

The overall percentage of laboratory-confirmed influenza cases was 29.2% (*n* = 359). The large majority of samples resulted in being positive for influenza A (H1N1)pdm09 virus (*n* = 345; 96.2%). The largest prevalence of laboratory-confirmed influenza cases among the total individuals with ILI monitored by the Sicilian RespiVirNet surveillance system was observed among subjects aged 5–14 years old (40.1%), females (29.8%) and subjects without comorbidities (30.6%) (Table 4).

During the 2023/2024 season, as of today, lower vaccination rates were reported among laboratory-confirmed influenza-positive cases compared to negative ones.

Specifically, lower vaccination rates were observed in all age groups (especially among ≤4 years old: 9.3 vs. 14.5; 5–14 years old: 9.6 vs. 18.8; ≥65 years old: 36.0 vs. 39.8), except for individuals aged 45–64 years (23.4 vs. 16.1) (Table 5).

Furthermore, a lower prevalence of vaccinated patients, among influenza-positive samples, was observed, particularly among females (19.1% vs. 12.9%) and in subjects with no reported comorbidities (Table 5).

Table 6 reports the test-negative VE estimates with their 95%CIs for the A(H1N1)pdm09 subtype. The overall adj-VE estimates against A(H1N1)pdm09 were 41.4% (95%CI: 10.5–61.6%). When stratifying by age groups, the adj-VEs showed protective values among children aged 7 months–14 years (37.9%; 95%CI: −0.7–61.7%) and among the elderly (52.7%; 95%CI: −38.0–83.8%), while the VE was not calculable among subjects aged 15–64 years due to the irrelevant number of vaccines administered in this age group.

## 4. Discussion

To date, the early phase of the virologic surveillance conducted in Sicily for the current influenza season has shown the exclusive circulation of influenza type A viruses, with a clear predominance of A(H1N1)pdm09 (96.1%), even higher than reported in the whole country (77.7%) and in other interim reports published in the Northern Hemisphere (Alberta, Canada >90%) [2,7].

Among 1230 surveilled subjects, pediatric patients prevailed (*n* = 798; 64.9%). These data are consistent with International and National influenza-like illness surveillance reports that yearly demonstrated a higher prevalence in the age classes >7 months–4 years and 5–14 years, and with the evidence provided during the previous seasons [1,16].

The higher risk of acquiring influenza-like illness during the first years of life could be related with a very low influenza vaccination coverage, commonly reported in Italy and in Sicily, although it has been increasing since the 2020/2021 season, when, for the very first time, the Italian Ministry of Health introduced the recommendation for children aged ≤6 years [10,20,21].

Specifically in the Sicilian population, less than 12.4% of pediatric population received an influenza vaccine during the 2023/2024 season, despite the introduction of the intranasal live attenuated vaccine in the vaccination offer, which is supposed to improve vaccination coverage in this target group [17].

Also, the influenza vaccination rates reported in the age classes 15–64 years old (5.4%), in the elderly (49.5%), and in subjects with comorbidities (32.1%) were similar to data reported during previous influenza seasons from the Ministry of Health and the Sicilian public health authorities, supporting the representativeness of the population enrolled in terms of vaccination adherence [10,20].

This mid-term influenza VE report showed an overall 41.4% reduction in laboratory-confirmed influenza cases in the Sicilian general population. To the best of our knowledge, this paper represents one of the most updated reports in Europe providing estimates on VE for influenza A(H1N1)pdm09 virus due to the prevalent circulation of this subtype during the 2023/2024 influenza season in the Northern Hemisphere [1,2,9].

In our sample, the limited number of laboratory-confirmed influenza-positive samples other than A(H1N1)pdm09 made it impossible to obtain estimates on VE against the other most common circulating influenza viruses. Similarly, the small number of laboratory-confirmed influenza cases in individuals vaccinated against influenza among those aged 15–64 years limited the VE estimates in this age group.

The mid-term VE surveillance data published in Alberta (Canada) reported a greater vaccine effectiveness for influenza A(H1N1)pdm09 (61%; 95% CIs: 58–64) and, probably due to the higher number of sample sizes collected, for A(H3N2) (49%; 95%CI: 28–63%) and B (75%; 95% CI: 58–85%) subtypes [2].

At the same time, VE estimates for A(H1N1)pdm09 influenza viruses among the Sicilian general population and the elderly were lower than the one reported during previous influenza seasons in Sicily and in the interim report conducted in Canada [2].

Comparing the current 2023/2024 season with previous ones, when A(H1N1)pdm09 largely circulated, an unexpected higher percentage of vaccinated subjects emerged among laboratory-confirmed influenza cases (12.5% vs. 5.8% and 6.0% documented in 2017/2018 and 2019/2020, respectively), even higher than those reported in 2018/2019 and 2016/2017 when the A(H3N2) subtype prevailed, which is notoriously subject to antigenic drift, egg-based, and other mutations, much more frequently than A(H1N1)pdm09 virus [22,23,24].

In Sicily, based on the trend observed between the 50th week of 2023 and the 1st week of 2024, the 2023/2024 seasonal influenza epidemic seems to have started to descend. Therefore, as far as can be approximated, the mid-season VE estimates from our network are expected to be consistent with data referring to the whole influenza season [25].

Some limits could affect the present mid-term analysis. First, similarly to other mid-term estimates, the present data could be just a bit different from those measured at the end of the influenza 2023/2024 epidemic. However, it should be noted that, in both the Italian and Sicilian surveillance systems, from the second week of 2024, the epidemic curve started a descending slope, suggesting that more than half of the total ILIs for this influenza season have been considered in our analysis. For this reason, the mid-season estimates from our network could be considered to be consistent with final VE data of the 2023/2024 influenza season [8,9,25].

Second, the different characteristics of circulating influenza viruses and influenza season duration across the Northern Hemisphere should be taken into account. In particular, the other mid-term report from Alberta (Canada) highlighted an earlier start of influenza circulation since the end of October–early November 2023, with an anticipated epidemic peak in comparison with the European figures, and this could have determined an overall VE different from that measured in other countries [1,2,8,9].

Third, it cannot be excluded that such lower VE estimates may have also been sustained by unexpected drifted A(H1N1)pdm09 viruses circulating during the current season. To this purpose, we are already working on full-genome sequencing of a representative number of 2023/2024 influenza strains to be phylogenetically compared with strains collected from other seasonal epidemics and, more specifically, to the strains included in the comparison of the recommended vaccines.

Lastly, the study was carried out on a limited sample of subjects enrolled throughout the regional sentinel system, and thus, similarly to other studies, it is not able to be fully representative of the whole population and could limit the external validity of the study. At the same time, the Sicilian RespiVirNet sentinel system requires the participation of GPs and FPs that are in charge of a representative sample of the general population (minimum 4% of general population at the national level; 4.8% of general population in Sicily during the 2023/2024 season). The number of GPs and FPs who participated in the present study could be considered adequate to increase the external validity of the study results and make estimates similar to “real-world” data observed in the general population. Finally, the limited sample size could further affect the strength of some of our VE estimates, especially for age groups and subtypes, for the small number of cases sustained by other type/subtypes such as A(H3N2) and B, and vaccinated subjects among laboratory-confirmed influenza cases aged 15–64 years.

## 5. Conclusions

The Sicilian mid-term VE estimates represent, to our knowledge, the first dataset on influenza vaccine effectiveness for the current season in Italy and Europe, and one of the first VE estimates for A(H1N1)pdm09 worldwide.

The observed VEs confirmed the protective role of vaccination in preventing laboratory-confirmed cases of influenza, especially among the elderly, that represent one of the categories at higher risk for influenza-correlated complications and mortality.

Moreover, our findings support the effectiveness of influenza vaccines against A(H1N1)pdm09, also during an influenza season, never seen before in Sicily, characterized by a higher prevalence of vaccinated subjects among laboratory-confirmed influenza cases.

Last but not least, our data support the need to implement phylogenetic analyses of the hemagglutinin gene of influenza A(H1N1) pdm09 viruses circulating during the 2023/2024 season, in comparison with the subtype recommended by the WHO for vaccine composition.

## Figures and Tables

**Figure 1 vaccines-12-00305-f001:**
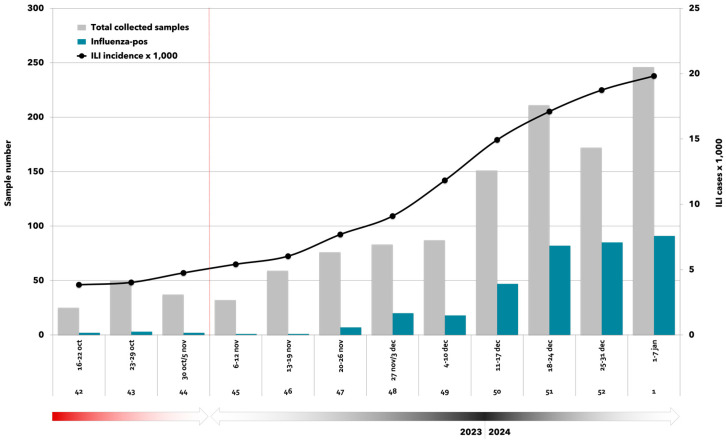
Weekly distribution of total collected samples (*n* = 1230), influenza laboratory-confirmed samples (*n* = 359), and ILI incidence per 1000 inhabitants [7]. RespiVirNet surveillance system and surveillance season 2023/2024 (16 October 2023–7 January 2024).

**Table 1 vaccines-12-00305-t001:** Influenza vaccines offered in Sicily since the beginning of the 2023/2024 influenza vaccination campaign: characteristics and recommendations for appropriate administration.

Influenza Vaccine Types Offered during the 2023/2024 Influenza Seasonal Vaccination Campaign in Sicily	Vaccine Characteristics and Recommendations for Different Target Groups of Population *
QIV-sd (Inactivated quadrivalent influenza vaccine standard dose)	Vaccine indicated to subjects aged 6 months and older without comorbidities and not allergic to any components of influenza egg-based vaccines.
2. QIV-cc (Cell culture-based inactivated quadrivalent influenza vaccine)	Vaccine indicated to subjects aged 25 months and older, especially if subjects are at high risk (affected by chronic comorbidities, healthcare professionals, pregnant women, or allergic to any component of influenza egg-based vaccines).
3. LAIV (Live attenuated quadrivalent influenza vaccine)	Vaccine indicated for subjects aged from 25 months to 17 years old (not recommended if immunocompromised, affected by severe and exacerbated asthma, or allergic to any components of influenza egg-based vaccines).
4. QIV-hd (High-dose inactivated quadrivalent influenza vaccine)	Vaccine indicated to subjects aged 60 years and older, especially if immunocompromised and resident in Long-Term Care Facilities.
5. QIV-a (Adjuvanted with MF59 inactivated quadrivalent influenza vaccine)	Vaccine indicated to subjects aged 65 years and older with or without comorbidities (not if immunocompromised or resident in Long-Term Care Facilities).

* In accordance with the Italian and Sicilian regional decrees on Influenza Control and Prevention.

**Table 2 vaccines-12-00305-t002:** Seasonal distribution of oropharyngeal samples, by influenza vaccination status, and prevalent influenza virus type/subtype circulating in Sicily. Seasons 2015/2016–2023/2024.

		Influenza-Positive Samples		Prevalent Seasonal Influenza Virus	
Surveillance Season	Total Collected Samples (*n*)	All Subjects *n* (%)	Vaccinated Subjects *n* (%)	Type/Subtype	% *
2015/2016	1414	652 (46.1)	25 (3.8)	A(H3N2); B	48.4; 39.9
2016/2017	1594	609 (38.2)	55 (9.0)	A(H3N2)	93.4
2017/2018	2075	1127 (54.3)	65 (5.8)	B; A(H1N1)pdm09	57.2; 42.1
2018/2019	2219	841 (37.9)	77 (9.2)	A(H3N2); A(H1N1)pdm09	62.2; 37.7
2019/2020	2494	1092 (43.8)	66 (6.0)	A(H1N1)pdm09; A(H3N2); B	40.5; 37.2; 22.3
2020/2021	28,730	0	-	-	-
2021/2022	5291	31 (0.6)	1 (3.2)	A(H3N2)	96.8
2022/2023	1338	431 (32.2)	13 (3.0)	A(H3N2)	83.7
2023/2024 **	1230	359 (29.2)	45 (12.5)	A(H1N1)pdm09	96.2

* Only prevalent type/subtype (≥20%) are reported. ** Samples collected at mid-term of season 2023/2024.

**Table 3 vaccines-12-00305-t003:** Total and laboratory-confirmed influenza-positive vaccinated subjects (with one of the influenza vaccines offered in Sicily during 2023/2024 season) among ILI subjects in Sicily. RespiVirNet surveillance system (period: 16 October 2023–7 January 2024).

Influenza Vaccine Type	Vaccinated Subjects	
	Total (*n*)	Influenza-Positive *n* (%)
QIV-sd (Inactivated quadrivalent influenza vaccine standard dose)	68	20 (29.4)
QIV-cc (Cell culture-based inactivated quadrivalent influenza vaccine)	9	1 (11.1)
LAIV (Live attenuated quadrivalent influenza vaccine)	67	17 (25.4)
QIV-hd (High-dose inactivated quadrivalent influenza vaccine)	13	3 (23.1)
QIV-a (MF59-adjuvanted inactivated quadrivalent influenza vaccine)	34	4 (11.8)
Overall subjects vaccinated	191	45 (23.6)

**Table 4 vaccines-12-00305-t004:** Characteristics of the 1230 individuals with ILI, according to influenza type/subtype detection. Sicilian RespiVirNet surveillance system: 16 October 2023–7 January 2024.

	Overall *n* = 1230	Laboratory-Confirmed Influenza Cases (*n* = 359; 29.2%)	Influenza Type/Subtypes, *n* (%)	
			Influenza A(H1N1)pdm09 (*n* = 345; 96.2%)	Influenza A(H3N2) (*n* = 14; 3.8%)
**Age groups, *n* (%)**				
≤4 years	461 (37.5)	129 (28.0)	125 (96.9)	4 (3.1)
5–14 years	337 (27.4)	135 (40.1)	129 (95.6)	6 (4.4)
15–24 years	34 (2.8)	7 (20.6)	7 (100)	0 (0)
25–44 years	85 (6.9)	16 (18.8)	14 (87.5)	2 (12.5)
45–64 years	165 (13.4)	47 (28.5)	47 (100)	0 (0)
≥65 years	148 (12.0)	25 (16.9)	23 (92.0)	2 (8.0)
**Sex, *n* (%)**				
Female	625 (50.8)	186 (29.8)	177 (95.2)	9 (4.8)
Male	605 (48.2)	173 (28.6)	168 (97.1)	5 (2.9)
**Comorbidities, *n* (%)**				
No	1055 (85.8)	323 (30.6)	309 (96.3)	12 (3.7)
At least one	175 (14.2)	36 (20.6)	34 (94.4)	2 (5.6)

**Table 5 vaccines-12-00305-t005:** Characteristics of the 1230 ILI subjects sampled by general practitioners and pediatricians between 16 October 2023 and 7 January 2024 in Sicily (Italy) according to influenza laboratory confirmation and influenza vaccination receipt.

Characteristics	Influenza-Positive *n* = 359 (29.2%)		Influenza-Negative *n* = 871 (70.8%)	
	Not Vaccinated	Vaccinated	Not Vaccinated	Vaccinated
**Age groups, *n* (%)**				
≤4 years	117 (90.7)	12 (9.3)	290 (85.5)	42 (14.5)
5–14 years	122 (90.4)	13 (9.6)	170 (81.2)	32 (18.8)
15–24 years	7 (100.0)	0	26 (96.2)	1 (3.8)
25–44 years	16 (100.0)	0	69 (100.0)	0
45–64 years	36 (76.6)	11 (23.4)	99 (83.9)	19 (16.1)
≥65 years	16 (64.0)	9 (36.0)	74 (60.2)	49 (39.8)
**Sex, *n* (%)**				
Female	162 (87.1)	24 (12.9)	355 (80.9)	84 (19.1)
Male	152 (87.9)	21 (12.1)	373 (86.4)	59 (13.6)
**Comorbidities, *n* (%)**				
No	289 (89.5)	34 (10.5)	633 (86.5)	99 (13.5)
At least one	25 (69.5)	11 (30.5)	95 (68.4)	44 (31.6)

**Table 6 vaccines-12-00305-t006:** Crude and adjusted test-negative vaccine effectiveness (adj-VE) influenza estimates with 95% confidence intervals (CIs) for influenza A(H1N1)pdm09 according to age groups.

Vaccine Effectiveness (VE) (Test-Negative)	Overall (*n*. 1230)		7 Months–14 Years Old (*n*. 798; 64.9%)		15–64 Years Old (*n*. 284; 23.1%)		≥65 Years Old (*n*. 148; 12.0%)	
Subtype	Crude VE	adj-VE *	Crude VE	adj-VE^*^	Crude VE	adj-VE *	Crude VE	adj-VE *
	(95% CIs)		(95% CIs)		(95% CIs)		(95% CIs)	
**Influenza A(H1N1)pdm09**	47.8 (20.9 to 65.5)	41.4 (10.5 to 61.6)	39.1 (1.5 to 62.4)	37.9 (−0.7 to 61.7)	N.C.**	N.C.**	52.5 (−38.3 to 83.7)	52.7 (−38.0 to 83.8)

* Vaccine effectiveness adjusted (adj-VE) by sex and at least one comorbidity. ** Not calculable

## Data Availability

Data not submitted for ethical restrictions and contained in the present study are available upon motivated request to the corresponding author.

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
