# Peer review of "Mid-Term Estimates of Influenza Vaccine Effectiveness against the A(H1N1)pdm09 Prevalent Circulating Subtype in the 2023/24 Season: Data from the Sicilian RespiVirNet Surveillance System"

_vaccines, 2024, doi:10.3390/vaccines12030305_

Round 1

Reviewer 1 Report

Comments and Suggestions for Authors

The manuscript "Mid-term estimates of influenza vaccine effectiveness against the A(H1N1)pdm09 prevalent circulating subtype in the 2023/24 season: data from the Sicilian RespiVirNet surveillance system" is a study to estimate the effectiveness of influenza vaccine using the data from the surveillance system. The design of the study is appropriate. It is a routine report of regional data so that the data can be referenced by readers in the field, but the overall novelty and scientific significance are relatively low, 

1. The method part will need to provide more details. Ideally, readers should be able to follow your method to repeat the study. For example, in lines 124-126, It needs to provide model and method details so others can use your method or at least cite the original paper of this method.  Please revise the method section to ensure the analysis is reproducible by others.

2. The analysis and interpretation of data should be cautious to avoid overstatement. For example, data in Table 3 only demonstrate with percentage, which may show some difference of VE among vaccines, but without significance analysis, you can not really state "lower 160 frequencies were found among subjects vaccinated with ...." as the small difference may be due to limited sampling. I would suggest doing the significance analysis for both Table 3 and Table 4.

3. table 5 should calculate the odd ratio to demonstrate if the vaccination protects people from infection.

3. figure 1 can be misleading as it looks like the study sampled all the cases, but it actually sampled ~1% of cases.  if you would show the sampling peak is consistent with the season peak, I would suggest using two panels; the top panel can be the sampling, and the bottom panel can show the season peak, ideally the epi curve. 

Comments on the Quality of English Language

Language is fine, please proofread and minor revise.

Author Response

General Comment: The manuscript "Mid-term estimates of influenza vaccine effectiveness against the A(H1N1)pdm09 prevalent circulating subtype in the 2023/24 season: data from the Sicilian RespiVirNet surveillance system" is a study to estimate the effectiveness of influenza vaccine using the data from the surveillance system. The design of the study is appropriate. It is a routine report of regional data so that the data can be referenced by readers in the field, but the overall novelty and scientific significance are relatively low. Language is fine, please proofread and minor revise.

A. We thank the reviewer for appreciating our manuscript. We agree the data are regional, but similar analysis are frequently regional (I.e. the Canadian Colleagues in Alberta) and also the novelty of this kind of work is limited to the seasonal interest of VE evaluation. The scientific significance is quite important because these researches could support the verification of the correct composition of influenza vaccines during the present season, in order to further improve that composition in the next seasons. Finally, thank you for appreciation of English language quality, a further proofreading was conducted through the manuscript in accordance with your suggestion.

The method part will need to provide more details. Ideally, readers should be able to follow your method to repeat the study. For example, in lines 124-126, It needs to provide model and method details so others can use your method or at least cite the original paper of this method.  Please revise the method section to ensure the analysis is reproducible by others.

A. We thank the reviewer for the useful comment. Also in accordance with similar suggestions of reviewer 2 we further improve the methods section in order to better explain the test negative analysis to other researchers and make it reproducible.

The analysis and interpretation of data should be cautious to avoid overstatement. For example, data in Table 3 only demonstrate with percentage, which may show some difference of VE among vaccines, but without significance analysis, you can not really state "lower 160 frequencies were found among subjects vaccinated with ...." as the small difference may be due to limited sampling. I would suggest doing the significance analysis for both Table 3 and Table 4.

A.We thank the reviewer for the comment. Similarly to table 2, Table 3 and 4 are merely descriptive tables. The two tables analyze the overall frequencies of laboratory confirmed influenza-positive vaccinated subjects (with one of the influenza vaccines offered in Sicily during 2023/2024 season) among the overall ILI subjects in Sicily (table 3) and the characteristics of the 1,230 individuals with ILI, according to influenza type/subtype detection (table 4).

The tables wants to evidence that there are no significant differences in accordance with vaccination type administered and in accordance with influenza type/subtype detected in terms of prevalence of infection and then we choose with other authors to not report not significant p-value.

In order to accomplish with your suggestion in the text the descriptions of the two tables were implemented and improved, underline the merely descriptive nature of the analysis.

Table 5 should calculate the odd ratio to demonstrate if the vaccination protects people from infection.

A. We thank the Reviewer for this comment. Nevertheless, it should be recalled that Table 5 aimed to preliminarily describe the overall frequencies of influenza-positive and influenza-negative subjects according to their flu vaccination status, in order to calculate “Crude and adjusted test-negative VE (adj-VE) influenza estimates”, as reported in Table 6.

This latter, in fact, included the test-negative VE estimates calculated exclusively for influenza A(H1N1)pdm09 and aggregated for some of the age-groups.

For Table 5, the odds ratios were calculated by stratum although, obviously, we were not able to obtain estimates for each age-group, due to the small (i.e. 45-64 uears old) or zero number of vaccinated individuals (i.e. 15-24 and 25-44 years old). We sincerely appreciated the suggestion but consider it inappropriate to report the Crude OR values into this context, in order to not confound the reader with a duplicate data

Figure 1 can be misleading as it looks like the study sampled all the cases, but it actually sampled ~1% of cases.  if you would show the sampling peak is consistent with the season peak, I would suggest using two panels; the top panel can be the sampling, and the bottom panel can show the season peak, ideally the epi curve. 

A. We agree with the Reviewer’s concern regarding the limited population-sampling, which necessarily constitutes a small number of total Sicilian influenza cases.

However, Figure 1 only depicts the trend of all collected samples in Sicily (n=1,230) and tested by our Reference laboratory. Moreover, according to Reviewer’s suggestion, the black curve illustrates the weekly incidence of ILI cases (“epi curve”) derived by the official data reported by the National Institute of Health, and demonstrated a similar trend. The reference [7] has been included in the description of Figure 1.

Reviewer 2 Report

Comments and Suggestions for Authors

INTRODUCTION

In the introduction  the authors state that  data were collected by “ sentinel general practi tioners (GPs) and family pediatricians”. Please indicate how GP and Pediatricians are selected to be part of sentinel system , Is it a random?  Are they  volunteers?

MATERIAL AND METHODS

When writing a paper, we should consider that the reader could be a person who is not expert in the topic or specialty. Due to this, negative test design, which is not often used, should be explained in detail.

In a test-negative design study, patients seeking medical care for acute respiratory illness are tested for the disease in question, in this case, the A(H1N1)pdm09 virus, a subtype of the influenza virus. Those who test positive for the virus are classified as cases, and those who test opposing serve as controls.

               This design allows researchers to compare the vaccination status of cases (those who tested positive for the virus) with that of controls (those who tested negative) to estimate the vaccine's effectiveness. The logic behind this design is that only including individuals who sought medical care reduces the bias related to healthcare-seeking behavior, as both cases and controls sought care for similar symptoms. This design is handy for evaluating the effectiveness of vaccines in real-world settings and has been widely used in influenza vaccine effectiveness studies.

In the specific study mentioned, the test-negative design was used to estimate the effectiveness of the seasonal influenza vaccine against the A(H1N1)pdm09 virus, which was the predominant subtype in Sicily during the study period, accounting for 96.2% of laboratory-confirmed influenza cases.

The reader should be able to understand a table , without readin the text of the paper. All acronyms used in tables should be explained. Eg in table 6, at the foot of the table it should be explained what VE (adj-VE) means (Probabliy is vaccine efficacy)

DISCUSSION

 Discuss how the collecting of data, using the sentinel system may affect the external validity of the study.

Author Response

In the introduction the authors state that data were collected by “sentinel general practitioners (GPs) and family pediatricians”. Please indicate how GP and Pediatricians are selected to be part of sentinel system. Is it a random?  Are they volunteers?

A. We thank the Reviewer for this comment. The Italian National Surveillance system of sentinel GPs and FPs is normally on voluntary basis. Specifically, in Sicilian Region, a central selection of the volunteers was annually conducted (substituting GPs of FPs that not worked well or retired with new ones) by the Regional Sections of the Scientific Societies of general practitioners (GPs) and family pediatricians. This point in accordance was specified in the text.

MATERIAL AND METHODS: When writing a paper, we should consider that the reader could be a person who is not expert in the topic or specialty. Due to this, negative test design, which is not often used, should be explained in detail.
In a test-negative design study, patients seeking medical care for acute respiratory illness are tested for the disease in question, in this case, the A(H1N1)pdm09 virus, a subtype of the influenza virus. Those who test positive for the virus are classified as cases, and those who test opposing serve as controls
This design allows researchers to compare the vaccination status of cases (those who tested positive for the virus) with that of controls (those who tested negative) to estimate the vaccine's effectiveness. The logic behind this design is that only including individuals who sought medical care reduces the bias related to healthcare-seeking behavior, as both cases and controls sought care for similar symptoms. This design is handy for evaluating the effectiveness of vaccines in real-world settings and has been widely used in influenza vaccine effectiveness studies.
In the specific study mentioned, the test-negative design was used to estimate the effectiveness of the seasonal influenza vaccine against the A(H1N1)pdm09 virus, which was the predominant subtype in Sicily during the study period, accounting for 96.2% of laboratory-confirmed influenza cases.
The reader should be able to understand a table , without readin the text of the paper. All acronyms used in tables should be explained. Eg in table 6, at the foot of the table it should be explained what VE (adj-VE) means (Probabliy is vaccine efficacy).

A. We thank the Reviewer for this comment and we fully agree with this suggestions. We correct the table 6 adding the foot of the table in accordance. Moreover, we better specify in the material and methods section the test-negative design methods in order to help also readers not expert in the topic or specialty and to also improve the reproducibility of the data (as suggested similarly by reviewer 1).

DISCUSSION: Discuss how the collecting of data, using the sentinel system may affect the external validity of the study.

A. We thank the Reviewer for this comment. External validity is the extent to which you can generalize the findings of a study to a broader context, in this case to general population. We believe that the data collection of the Italian (and the Sicilian RespiVirNet system), accounting for a representative sample of general population (average 4% at National Level, in Sicily 4.6%) could be considered the best way to increase the external validity of the study results to general population and make estimates similar to “real world” data.
We add a specific sentence to the limitation section in Discussion.

Reviewer 3 Report

Comments and Suggestions for Authors

The manuscript titled "Mid-term estimates of influenza vaccine effectiveness against the A(H1N1)pdm09 prevalent circulating subtype in the 2023/24  season: data from the Sicilian RespiVirNet surveillance system" provides stats on influenza positive cases in Italy during the current influenza season. Overall, the manuscript is well organized. The tables are easy to read, methods are well-described, and the discussion support the results as well as a thorough discussion of study limitations. This report is a valuable tool to understand the current impacts of influenza in both unvaccinated and vaccinated populations. I have only one suggestion to offer. While this report is of high value in the surveillance of seasonal influenza, the introduction could be strengthened by emphasizing the important of such data and reporting. A discussion on how this data can help determine efficacy of vaccination programs or track trends in infection cases would help bolster the significance of the report. 

Author Response

General Comment: The manuscript titled "Mid-term estimates of influenza vaccine effectiveness against the A(H1N1)pdm09 prevalent circulating subtype in the 2023/24  season: data from the Sicilian RespiVirNet surveillance system" provides stats on influenza positive cases in Italy during the current influenza season. Overall, the manuscript is well organized. The tables are easy to read, methods are well-described, and the discussion support the results as well as a thorough discussion of study limitations. This report is a valuable tool to understand the current impacts of influenza in both unvaccinated and vaccinated populations.

I have only one suggestion to offer. While this report is of high value in the surveillance of seasonal influenza, the introduction could be strengthened by emphasizing the important of such data and reporting.

A discussion on how this data can help determine efficacy of vaccination programs or track trends in infection cases would help bolster the significance of the report. 

A. Firstly, we want to thank the reviewer for appreciating our manuscript, the organization, the analysis and the data presentation. In accordance with their useful suggestion we strengthen the introduction section on the seasonal VE surveillance against influenza and on how this data can help determine efficacy of vaccination programs or track trends in infection cases.

Reviewer 4 Report

Comments and Suggestions for Authors

Review
Dear Authors,
The paper is very interesting and pertinent.
The subject is totally topical and that is why I think it is necessary to publish it.
Technically it is correct and simple, but I would like to make a comment.
In the Data Analysis section it is said that odd ratios are calculated by logistic regression.
It would be very explanatory to say also how the EV is calculated from these odd ratios. Table 6 is not well understood.
Otherwise, the paper is correct and I have nothing more to say.

Author Response

General Comment: Dear Authors,

The paper is very interesting and pertinent.

The subject is totally topical and that is why I think it is necessary to publish it.

Technically it is correct and simple, but I would like to make a comment.

In the Data Analysis section it is said that odd ratios are calculated by logistic regression.

It would be very explanatory to say also how the EV is calculated from these odd ratios. Table 6 is not well understood.

Otherwise, the paper is correct and I have nothing more to say.

A. Dear Reviewer, firstly, we want to thank you for appreciating our manuscript, the organization, the analysis and the data presentation. In accordance with your useful suggestions we improved Table 6 explaining what adj (adjusted) ORs and Crude ORs for the VE calculated values means (also in accordance with similar request raised by reviewer 1). We also add a specific section in material and methods that better clarificate the test negative design for the calculation of VE.